# Influence of Cognitive Factors on Adherence to Social Distancing and the Use of Masks during the COVID-19 Pandemic by Young Adults: A Systematic Review

**Marina Almeida-Silva [1,2], Graça Andrade [1,*], Tamara Luis [3], Margarida Santos [3,4] and Ana Grilo [1,4]**

1   H&TRC—Health & Technology Research Center, ESTeSL—Escola Superior de Tecnologia da Saúde, Instituto Politécnico de Lisboa, Estr. de Benfica 529, 1549-020 Lisboa, Portugal; marina.silva@estesl.ipl.pt (M.A.-S.); ana.grilo@estesl.ipl.pt (A.G.)
2   OSEAN-Outermost Regions Sustainable Ecosystem for Entrepreneurship and Innovation, 9000-082 Funchal, Portugal
3   Escola Superior de Tecnologia da Saúde de Lisboa, Instituto Politécnico de Lisboa, Av. Dom João II Lote 4.69 01, 1990-096 Lisboa, Portugal; tamaraluis97@gmail.com (T.L.); margarida.santos@estesl.ipl.pt (M.S.)
4   Research Center for Psychological Science, Faculdade de Psicologia, Alameda da Universidade, 1649-013 Lisboa, Portugal
*   Correspondence: mgandrade@estesl.ipl.pt

**Abstract:** Social distancing and the use of masks are crucial to prevent the spread of SARS-COV-2. Knowledge of the determinants of this behavior is essential to promote effective communication with the public in future public health crises that require mass public compliance with preventive behaviors. This systematic review focused on scientific evidence related to cognitive factors that underlie the intention of young adults' intention to adhere to preventive social behavior (distancing and/or the use of facial masks) against COVID-19. A systematic literature search on the electronic database, PubMed, Scopus, Web of Science, and EBSCO was performed in December 2022 according to Preferred Reporting Items for Systematic Reviews and Meta-Analysis guidelines. The PEO (Population: young adults, Exposure: COVID-19, and Outcome: cognitive factors that underlie the intention of young adults to adhere to social distancing and/or the use of facial masks) was developed to identify search terms and inclusion/exclusion criteria. Eight studies met the eligibility criteria. None of the studies were seriously flawed according to the quality assessment, and they were considered to have a low risk of bias for selection. Several cognitive determinants emerged in the analysis. For both social distancing and the use of masks, the most relevant factors related to adherence include risk perception and perceived severity, the moral value of fairness, social responsibility, trust in the government, respect for authority, and the quality of institutional communication. Adherence to social distancing was found related to self-efficacy. These results reinforce social cognitive models showing the relevance of cognitions to adherence behavior, and highlight the responsibility of official institutions in the development of contexts and in adapting the communication for the effective promotion of adherence to the recommendations they launch.

**Keywords:** COVID-19; social distance; mask; adherence; cognitive factors

## 1. Introduction

The COVID-19 epidemic was declared by the World Health Organization (WHO) as a public health emergency of international concern on 30 January 2020 (WHO 2020). Coronaviruses are a large family of viruses that cause illnesses ranging from the common cold to more severe diseases (Isaifan 2020).

The awareness of the rapid progression of contamination by SARS-CoV-2, together with the inexistence of a vaccine at the beginning of the pandemic, led the WHO to recommend the use of other measures to prevent contagion, namely face masks, social

distancing, maintaining good hand hygiene, avoiding direct hand contact with eyes, mouth, or nose, and covering the nose and mouth when sneezing or coughing with tissues or with a bent elbow (WHO 2022). The predominant way for transmission of COVID-19 is through the air, mainly by droplets that are produced when people cough or sneeze or even when talking (Cevik et al. 2020). As with SARS-CoV-2, there is evidence that droplets are the main way that most respiratory pathogens spread (e.g., influenza, meningococcus) (Jefferson et al. 2009). Therefore, physical distancing and the use of masks will be recommended when trying to decrease the risk of spread of other potential pathogenic similar diseases.

These measures, like the use of masks and social distancing, are human behaviors. According to social cognitive models (e.g., Social Cognitive Theory (SCT), Health Belief Model (HBM), and Theory of Planned Behavior) adherence to any health-related behavior involves a decision process (Ronis 1992; Misra and Kaster 2012). Health beliefs are what people think about their health and health events, the cause of their illness, and ways to overcome or prevent an illness. Evidence shows that health beliefs are essential components of health attitudes, decisions, and behaviors (Misra and Kaster 2012).

The Social Cognitive Theory (SCT) postulates that learning occurs in a social context with a reciprocal and dynamic interaction among person, environment, and behavior (Bandura 2001). Most cognitive health models encompass central concepts of SCT (Bandura 1998), namely self-efficacy, outcome expectations, and goals. Perceived self-efficacy refers to the "beliefs in one's capability of organize and execute the courses of action required to produce given levels of attainments" (Bandura 1998, p. 624) and involves the perception of control over health behaviors and biological processes. Although personal and environmental barriers to change are considered in the SCT, they are perceived as an integral part of the perception of self-efficacy assessment. Outcome expectations (positive or negative) are most important in deciding to adopt (or not) a particular health behavior. These expectations consider past experiences and include the behavior change's physical, social, and self-evaluative effects. Therefore, people adopt behaviors that they believe have a significant positive result and disregard others perceived as not having positive consequences. SCT explains how people adjust their behavior through reinforcement and control to achieve goal-directed behaviors that can be kept over time. These goals are a component of the self-motivation process and can reinforce and guide behavior change.

According to the Health Belief Model (HBM), individual beliefs will predict the possibility of adopting a preventive behavior or action (Rosenstock 2000). The HBM comprises five dimensions of health beliefs: health motivation, perceived susceptibility and severity of health threat, perceived benefits, barriers of the protective behavior, and self-efficacy to engage in this behavior (Rosenstock 2000). In this framework, for behavioral change to succeed, people must perceive themselves as threatened by their current behavioral patterns, be vulnerable to the health threat that is perceived as severe, believe that a specific change will result in a valued outcome at an acceptable cost, and feel competent to implement the change.

The Theory of Planned Behavior (TPB) focuses on relationships between attitudes, intentions, and behaviors (Ajzen and Driver 1991; Fishbein 1967). A central factor in the theory TPB is the individual's intention to perform a given behavior. As a rule, the stronger the intention to engage in a behavior, the more likely it should be its performance (Ajzen and Driver 1991, p. 181). This model postulates that, attitudes, subjective norm, and perceived control are conceptually independent determinants of intention (Fishbein and Ajzen 1975). Attitude toward a specific behavior reflects the degree to which a person has a favorable / unfavorable appraisal of a specific behavior, and entails a consideration of the outcomes of performing the behavior in question (Ajzen 1992). Therefore, a behavior attitude is determined by a person's beliefs about that behavior and the results that the behavior will generate (Ajzen and Driver 1991). Subjective norm addresses the pressure that the individual feels from those around them, or their social nucleus. It refers to a person's beliefs about whether peers and people of importance to the person think he or

she should engage in the behavior. It is understood that this social influence can significantly influence the adoption of a specific behavior (Fishbein and Ajzen 1975). Perceived control refers to the perceived ease or difficulty of performing the behavior and it is assumed to reflect experience as well as anticipated impediments and obstacles (Ajzen 1991). This belief is influenced either by past events or by the person's beliefs regarding possible obstacles in the performance of the behavior (Fishbein and Ajzen 1975).

There is evidence that other cognitive factors influence adherence to norms and rules. Among those factors, the role of "personal values" has been addressed during COVID-19, mainly: conservation, self-enhancement, openness-to-change values (Bonetto et al. 2021; Daniel et al. 2022; Potocan and Nedelko 2023; Vecchione 2022), self-transcendence (Daniel et al. 2022; Vecchione 2022), religious belief, government satisfaction, and individual freedom (Lyu et al. 2022). Also "age" has been studied as a determinant of adherence with COVID-19 public health preventive measures, with young adults identified as a group with low adherence to COVID-19 protective behaviors (WHO 2022).

However, despite the recognition of the relevance of cognitions in adherence behavior, there is a lack of systematized knowledge on applying cognitive models to understand adherence to individual protection recommendations, namely social distancing and using masks in young adults during the COVID-19 pandemic.

In previous centuries, pandemics were frequent. Alerting for the risk of new health pandemics shortly, WHO launched, in 2023, a new initiative called Preparedness and Resilience for Emerging Threats Initiative to guide countries in pandemic planning. This initiative urges governments to develop timely measures to prevent contamination in potentially epidemic health contexts. These measures necessarily include the development of strategies to promote adherence to healthy behaviors and preventive recommendations for specific groups such young adults (18–35 years old), as defined by Levinson (1978).

This study aims to systematize the scientific evidence related to cognitive factors that underlie the intention of young adults' intention to adhere to preventive social behavior (distancing and/or the use of facial masks) against COVID-19. We believe that the results of our study can contribute to the design of tailored communication and/or interventions that promote adherence to measures to prevent the transmission of new pandemics by young adults, thus responding to the call of the WHO.

## 2. Methods

### 2.1. Design

A systematic literature review of quantitative studies was performed to identify the cognitive determinants of adherence to social distance and using masks during the COVID-19 pandemic in young adults. The review was performed using the Preferred Reporting Items for Systematic Reviews and Meta-analysis (PRISMA), which encompassed three phases: Identification, Screening, and Included (Page et al. 2021). The PROSPERO registration number ID is CRD42023405658. The protocol is also available at the following link: https://www.crd.york.ac.uk/prospero/display_record.php?ID=CRD42023405658, accessed on 21 December 2023.

The PEO (Population, Exposure, and Outcome) statement was developed for this systematic review to identify search terms and inclusion/exclusion criteria (Table 1).

**Table 1.** PEO criteria for inclusion and exclusion in the systematic review.

| | |
|---|---|
| P | Young Adults |
| E | COVID-19 |
| O | Cognitive factors that underlie the intention of young adults to adhere to preventive behavior (social distancing and/or the use of facial mask) |

### 2.2. Search Strategy

Publications that describe original quantitative research were retrieved via electronic database searches of PubMed, Scopus, Web of Science, and Psychology and Behavioral Sciences Collection from EBSCO in December 2022. The keywords used with the Boolean operators AND and OR are presented in Table 2. No keywords related to the age group were included. This decision allowed the authors to make this selection, enabling more detailed scrutiny of specific data on 18–35 years (inclusion criteria).

**Table 2.** Detailed description of the keywords used in each scientific database.

| Databases | Keywords |
|---|---|
| PubMed | 1#<br>((“COVID*” OR “SARS*”) AND (“Masks”[Mesh] OR “mask use”) AND (“adherence” OR “compliance”) AND (“Risk Perception” OR vulnerability OR “self efficacy”[MeSH Terms] OR “self efficacy” OR “cost” OR “resources” OR “habits”[MeSH Terms] OR “environmental constrains” OR “barriers” OR “obstacles” OR “planning” OR “action planning” OR “coping planning” OR “social pressure” OR “social influence”)) AND (2019/1/1:2022/12/31[pdat]) |
| | 2#<br>((“COVID*” OR “SARS*”) AND (“Masks”[Mesh] OR “mask use”) AND (“adherence” OR “compliance”) AND (“Risk Perception” OR vulnerability OR “self efficacy”[MeSH Terms] OR “self efficacy” OR “cost” OR “resources” OR “habits”[MeSH Terms] OR “environmental constrains” OR “barriers” OR “obstacles” OR “planning” OR “action planning” OR “coping planning” OR “social pressure” OR “social influence”)) AND (2019/1/1:2022/12/31[pdat]) |
| | 3#<br>((“COVID*” OR “SARS*”) AND (“Masks”[Mesh] OR “mask use”) AND (“Health Belief Model” OR “Theory of Planned Behaviour” OR “Health Action Process Approach”)) AND (2019/1/1:2022/12/31[pdat]) |
| | 4#<br>((“COVID*” OR “SARS*”) AND (“Physical Distancing”[Mesh] OR “social distance”) AND (“Health Belief Model” OR “Theory of Planned Behaviour” OR “Health Action Process Approach”)) AND (2019/1/1:2022/12/31[pdat]) |
| Web of Science | 1#<br>ALL = (“COVID-19” OR “SARS-CoV2”) AND ALL = (“mask*”) AND ALL = (“adherence” OR “compliance”) AND ALL=(“risk perception” OR vulnerability OR “self efficacy” OR “self efficacy” OR “cost” OR “resources” OR “habits” OR “environmental constrains” OR “barriers” OR “obstacles” OR “planning” OR “action planning” OR “copings planning” OR “social pressure” OR “social influence”) |
| | 2#<br>ALL = (“COVID-19” OR “SARS-CoV2”) AND ALL = (“physical distanc*” OR “social distanc*”) AND ALL = (“adherence” OR “compliance”) AND ALL = (“risk perception” OR vulnerability OR “self efficacy” OR “self efficacy” OR “cost” OR “resources” OR “habits” OR “environmental constrains” OR “barriers” OR “obstacles” OR “planning” OR “action planning” OR “copings planning” OR “social pressure” OR “social influence”) |
| | 3#<br>ALL = (“COVID-19” OR “SARS-CoV2”) AND ALL = (“mask*”) AND ALL = (“health belief model” OR “theory of planned behaviour” OR “health action process approach”) |
| | #4<br>ALL = (“COVID-19” OR “SARS-CoV2”) AND ALL = (“physical distanc*” OR “social distanc*”) AND ALL = (“health belief model” OR “theory of planned behaviour” OR “health action process approach”) |
| Scopus | #1 |

| | |
|---|---|
| | ALL ("COVID-19" OR "SARS-CoV2") AND ALL ("mask*") AND ALL ("adherence" OR "compliance") AND ALL ("risk perception" OR "vulnerability" OR "self efficacy" OR "cost" OR "resources" OR "habits" OR "environmental constrains" OR "barriers" OR "obstacles" OR "planning" OR "action planning" OR "coping* planning" OR "social pressure" OR "social influence") AND PUBYEAR > 2019 AND PUBYEAR < 2023 |
| | #2 |
| | ALL ("COVID-19" OR "SARS-CoV2") AND ALL ("mask*") AND ALL ("adherence" OR "compliance") AND ALL ("risk perception" OR "vulnerability" OR "self efficacy" OR "cost" OR "resources" OR "habits" OR "environmental constrains" OR "barriers" OR "obstacles" OR "planning" OR "action planning" OR "coping* planning" OR "social pressure" OR "social influence") AND PUBYEAR > 2019 AND PUBYEAR < 2023 |
| | #3 |
| | ALL ("COVID*" OR "SARS*") AND ALL ("Mask*") AND ALL ("health belief model" OR "theory of planned behaviour" OR "health action process approach") AND > 2019 AND PUBYEAR < 2023 |
| | 4# |
| | ALL ("COVID*" OR "SARS*") AND ALL ("physical distanc*" OR "social distanc*") AND ALL ("health belief model" OR "theory of planned behaviour" OR "health action process approach") AND PUBYEAR > 2019 AND PUBYEAR < 2023 |
| EBSCO (Psychology and behavioral sciences collection) | 1. ("COVID-19" OR "SARS-CoV2") AND ("Masks" OR "mask use") AND (adherence OR compliance)<br>2. ("COVID-19 OR "SARS-CoV2") AND ALL=("Physical Distancing" OR "social distancing") AND ALL=("adherence" OR "compliance") |

The Rayyan software (Ouzzani et al. 2016) was used during the screening phase. Four authors in pairs (MAS and GA + MS and AG) first screened all abstracts and titles from the search to eliminate irrelevant studies. Then, the authors screened full-text articles and made final eligibility decisions based on inclusion/exclusion criteria (Table 3). In both phases, disagreements were resolved by discussion and consensus.

**Table 3.** Inclusion and exclusion criteria.

| Inclusion Criteria | Exclusion Criteria |
|---|---|
| - European studies<br>- Canadian studies<br>- North American studies<br>- Quantitative studies<br>- Young adults (18–35 years old)<br>- Articles published from January 2020 to December 2022 | - Studies focusing health professionals<br>- Studies with parents as informants (by proxy)<br>- Review and meta-analysis studies<br>- Specific groups, namely: pregnant women, patients |

### 2.3. Quality Appraisal

For Quality assessment, the JBI Critical Appraisal Checklist for Analytical Cross-Sectional Study, which encompasses eight criteria assessments (Moola et al. 2020), was used. This tool is the most used in the quality assessment of analytical cross-sectional studies (Ma et al. 2020). The only cohort study was assessed by the quality assessment based on JBI Critical Appraisal Checklist for Cohort Studies. Two authors (MAS, GA) independently evaluated all articles as "yes", "no", "unclear", and "not applicable" in the dimensions proposed on JBI tools. Consensus resolved disagreements, and a third reviewer (MS) was available to arbitrate any unresolved issues. After discussing the ratings and resolving any discrepancy, the global rating for each of the selected articles was obtained by dividing the sum of ratings given ("No" = 1; "Unclear" = 2; "Yes" = 3) for the number of dimensions.

According to the quality assessment, none of the eight studies were considered seriously flawed. The eight included studies scored between 10 and 21 in the JBI Checklist (see Supplementary Material: Figure S1 and Table S1). All studies were considered to have a low risk of selection bias (Figure 1). The only cohort study (Nivette et al. 2021) obtained a score of 19 in a range between 11 and 33. The authors decided not to apply any cut-off point, accepting all assessed articles.

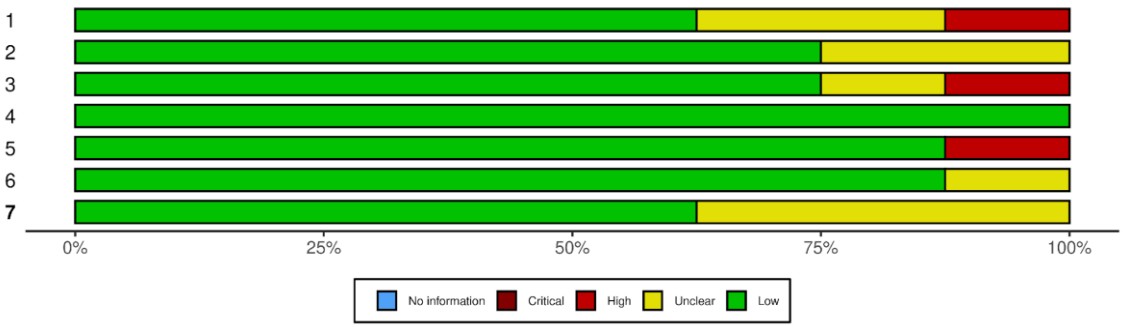

**Figure 1.** Summary of the critical appraisal based on JBI checklist. The numbers 1 to 7 are the assessed cross-sectional studies.

### 2.4. Data Extraction

Two reviewers (TL and MAS) extracted the data from all articles independently. For each study, information was extracted regarding the authors, publication date, country, goals, sample, study design, measures, and main results (Table 4).

**Table 4.** Sum up of the data provided by the analyzed sample.

| Ref. | Goals | Sample/Country | Study Design | Measures | Main Results |
|---|---|---|---|---|---|
| (Nivette et al. 2021) | To describe patterns of non-compliance with COVID-19 related public health measures in young adults and identify which characteristics increase the risk of non-compliance.<br><br>Characteristics under study: (a) prior social and psychological risk factors (weak social bonds, active social lifestyle, attitudes towards the law and police, deviant peers, and antisocial behavior and, dispositional factors) and (b) present attitudes to COVID prevention measures (risk perception, trust in government, and information seeking). | N = 737 Age—22 years old (assessments at 15, 17, 20 and 22 years old) Switzerland. | Prospective-longitudinal cohort study | Questionnaire with E13 questions about protective behaviors, reflecting national and international recommendations (Center of Disease Control; Federal Office of Public Health in Switzerland; WHO 2020).<br><br>Wearing a mask was exclude from analyses | Non-adherence to social distancing measures was associated factors: low police legitimacy [police performance, fairness and confidence in police effectiveness] low self-control and low general trust (this factor sought to capture the perception whether people can be trusted and help others).<br><br>Low trust in government was the only facto with significative association with social distancing. |
| (Müller and Rau 2020) | To analyze associations whether social responsibility is associated with higher social compliance with COVID preventive measures; how social responsibility and economic preferences shape people's perceptions of the crisis; how the three economic preferences (risk, time, trust) predict citizens' social compliance with political measures in the Corona crisis. | N = 185 University students with mean age of 22, 86 years 52% female Germany. | Cross-sectional | Two blocks of questions:<br><br>(1) general preferences, on: risk tolerance, time preferences, generalized trust, trustworthiness, and honesty.<br><br>(2) contextual questions on compliance in the COVID-19 time and about subjects' perception of the crisis. | Neither general trust nor trustworthiness, (i.e., trust in interpersonal relationships) are predictive for social compliance with social distancing. Participants who reveal a high degree of social responsibility tend to be more compliant with respect to staying at home and avoiding crowds during the crisis.<br>Less risk-tolerant citizens are prone to a greater perceived threat of Corona than more risk-tolerant individuals. Risk tolerance is predictive of some behaviors under COVID-19. Risk-tolerant citizens are less likely to avoid crowds. Participants with an above-median risk-tolerance are less likely to increase staying home and less often avoid crowds. Time preferences reflect the subject's impatience and suggest that more patient individuals are more likely to stay at home and avoid crowds. |



| | | | | | |
|---|---|---|---|---|---|
| (Barrett and Cheung 2021) | To identify (1) the socio-cognitive perceptions towards hand hygiene and social distancing and (2) which determinants (such as knowledge and socio-cognitive perceptions) explain hand hygiene and social distancing. | N = 293<br>Age: Range from 18–52 y<br>Median age: 22 years<br>Subjects in the range 18–25 years) N = 215<br>65.2% Female<br>UK. | Cross-sectional | Online Survey questionnaire: demographics, knowledge of the disease and effectiveness of the protective measures, risk perception, socio-cognitive perceptions (e.g., attitude, social support, and self-efficacy), habit, time factors and trust, as well as the hand hygiene and social distancing behaviors. | A significant positive correlations were found between social distancing behavior and advantages perception, social support, self-efficacy for social distancing, aspects of trust in the restrictive policies, and self-efficacy for infection avoidance.<br>Risk perception components, social support/social norms, knowledge of the disease or effectiveness of the specific performance of hygiene behaviors or social distancing and disadvantages of this behaviors did not separately contribute to the model.<br>Self-efficacy was a major predictor for social distancing behavior, followed by confidence in restrictive measures during COVID, and perceived advantages. |
| (Luo et al. 2021) | To explore age disparities in the perceived severity of COVID-19 and in the adoption of preventive measures.<br>Investigate how the perceived severity of the virus influences the generational gap in preventive behaviors.[1] | N(total) = 1843<br>Age—Range from 18 years old to >55<br>N(18–24 years) = 191<br>N(25–39 years) = 521<br>56.7% Female<br>USA. | Cross-Sectional | Survey questionnaire in 3 domains: perceived severity; preventive actions (mainly social distance behavior and also use of mask); information. | Younger show less preventive behaviors than older people, with no differences found at ages 18–24 and 25–39.<br>Younger (18–24) have a lower perception of risk to COVID-19.<br>Perceived severity of COVID-19 is higher at ages 25–39 than at ages 18–24.<br>The difference in preventive actions between the 18–24 and 50 age groups tends to decrease as the level of perceived severity increases difference of 0.15).<br>Information was a determinant of perceived severity (no specific results for 18–24 Y group). |

| | | | | |
|---|---|---|---|---|
| (White et al. 2022) | To identify HBM constructs related to mask wearing. | Representative sample of US adults from 18 to 49 years N(total) = 474 N(>30 years = 842) USA. | Cross-Sectional | Online survey. Respondents answer in a Likert scale to questions assessing HBM constructs: perceived susceptibility and perceived severity of COVID-19; face mask perceived benefits, barriers and efficacy. Face mask behavior was assessed by 2 questions (5 point Likert scale) about the frequency of mask wearing (when around people who do not live in their household; in public when not able to stay 6 ft away from others). | Perceived COVID-19 severity, masking benefits, and efficacy were positively associated with masking behavior.

Perceived masking barriers were negatively associated with masking behavior.

Susceptibility and cues to action were not significantly associated with participants' masking behavior. |

| (Coroiu et al. 2020) | This study has three aims:<br>1. To describe rates of motivations (barriers and facilitators) for social distancing.<br>2. To describe rates of adherence to social distancing recommendations.<br>3. To investigate the relationship between socio-demographic characteristics, psychological variables, and motivations for social distancing and adherence to social distancing recommendations. | N = 2013<br>N(18–24 years) = 231<br>N(25–44 years) = 922<br>84% female<br>Europe and North America. | Cross-Sectional | Recruitment and data collection were conducted online using the Qualtrics platform.<br><br>Distributed via snowball.<br><br>Predictor variables:<br>- Sociodemographic and medical information.<br>- MacArthur scale of subjective social status scale.<br>- Health literacy scale.<br>- Belief in conspiracy theories scale.<br>- Pro-social behavioral intentions scale.<br>- Patient health questionnaire-4 (PHQ-4).<br>- Motivations for social distancing and social distancing behaviors. | The more relevant facilitators for adherence to social distancing recommendations include:<br>- Wanting to protect the self.<br>- Feeling a responsibility to protect the community.<br>- Being able to work/study remotely.<br><br>The more relevant barriers for adherence to social distancing recommendations include:<br>- Having friends.<br>- Family who needed help with running errands.<br>- Socializing in order to avoid feeling lonely.<br><br>More relevant predictors of social distance were: motivation for protection (self and others) and prosocial attitudes |

| | | | | | |
|---|---|---|---|---|---|
| (Hsing et al. 2021) | To compare handwashing and social distancing practices in different countries and evaluate practice predictors using the health belief model (HBM). | N(total) = 7016<br>N(18–34 years) = 700<br>N(25–34 years) = 1670<br>55% female<br>United States, Mexico, Hong Kong (China),<br>and Taiwan | Cross-Sectional | International open survey through the following social media platforms: Facebook, Instagram, Line, and Twitter.<br><br>Assessed action/Behavior: Social distancing and Handwashing.<br><br>Individual Beliefs:<br>- Perceived susceptibility to acquiring COVID-19.<br>- Perceived severity of COVID-19.<br>- Perceived benefits of local government measures.<br>- Perceived barriers to adhering to recommended measures.<br>- Self-efficacy in carrying out preventive measures. | Social distancing was positively associated with perceived severity.<br>Perceived susceptibility:<br>- 54.9% felt they were likely to be infected with COVID-19.<br>Perceived severity:<br>- 28.6% were not afraid of the COVID-19.<br>Perceived benefits:<br>- 68.4% believed that the government measures in place were appropriate or essential.<br>Perceived barriers:<br>- 52.7% of individuals who perceived difficulty in obtaining face masks.<br>Self-Efficacy:<br>- 88.5% s were confident or very confident in their ability to carry out preventative measures. |

| | | | | |
|---|---|---|---|---|
| | | | Recruitment and data collection were conducted online using the Qualtrics platform. | |
| (Hunt et al. 2022) | To examine the correlates of core values and social influence on mask non-compliance in undergraduates at a selective American university. | N = 113 university students 61% female Mean age = 19.9 USA | Cross-Sectional | Variables that were measured along Mask-Wearing:<br><br>- Fear of COVID-19.<br>- Heath anxiety (i.e., fear of becoming ill).<br>- Political affiliation (i.e., conservative//liberal).<br>- Moral foundations (moral core values, e.g., fairness, respect for others, respect for authority). | - Fear of COVID-19 was robustly positively correlated with adherence.<br>- To be political conservative was robustly associated with mask non-compliance (r = −0.459, *p* < 0.001). Fairness was robustly associated with mask compliance (r = 0.43, *p* < 0.001).<br>- Valuing group loyalty and respect for authority were correlated with mask non-compliance (r = −0.35 and r = −0.40, both *p* < 0.001, respectively). |

[1] The behaviors involved were: clean hands often, wear a face mask outsider, limit outdoor activities, avoid attending mass gathering., keep social distance with others, and avoid close contact with people who are sick.

## 3. Results

### 3.1. Studies Selection

The progress through the stages of the systematic review is summarized in Figure 2.

**Figure 2.** Search and study selection PRISMA flow diagram.

From the initial search, 3171 articles were obtained. Of those, 1042 were removed were withdrawn due to duplication were duplicates. Such high numbers were due to the use of four different search platforms. During the next phase, 1946 articles were excluded after titles and abstracts of all search results were independently screened for relevance.

Upon exclusion of irrelevant articles, the remaining 183 full-text articles were accessed and screened for eligibility, and 175 articles were excluded, with the most frequently reported exclusion criteria being: age (including unspecified ages, age range outside 18–35 years old, no specific results for this age group), the outputs were out of scope (i.e., did not meet the keywords specified in the search), not meeting the criteria of geographic location, or no data available.

Overall, eight articles were included.

### 3.2. Characteristics of Included Studies

Table 4 reports the characteristics of the eight included studies. Of the eight studies, three were conducted in the United States of America (Luo et al. 2021; White et al. 2022; Hunt et al. 2022), one in Switzerland (Nivette et al. 2021), one in Germany (Müller and Rau 2020), another one in the United Kingdom (Barrett and Cheung 2021). Two were international studies, one involving several European countries and North America (Coroiu et al. 2020), and the other was carried out with the United States, Mexico, China, and Taiwan population. All were cross-sectional studies, except one with a longitudinal design (Nivette et al. 2021). All studies used online questionnaires.

Of the eight studies analyzed, the age groups range from 18 to 75 years old, with a total of 6.215 subjects between 18 and 35 years. Thus, only specific data relating to the age group in the inclusion criteria were considered. In most studies, more women than men participated (Hunt et al. 2022; Barrett and Cheung 2021; Hsing et al. 2021; Luo et al. 2021; Müller and Rau 2020; Coroiu et al. 2020).

### 3.3. Outcomes

The studies under analysis evaluate the influence of cognitive factors on adherence to social distancing (Hsing et al. 2021; Barrett and Cheung 2021 Coroiu et al. 2020; Müller and Rau 2020; Nivette et al. 2021), use of mask (Hunt et al. 2022; White et al. 2022), and several preventive behaviors mainly related with social distancing (Luo et al. 2021). The cognitive factors considered in the studies were: fear/severity/risk of COVID-19 (Hunt et al. 2022; White et al. 2022; Barrett and Cheung 2021; Hsing et al. 2021; Luo et al. 2021; Müller and Rau 2020) and susceptibility to be infected (White et al. 2022; Hsing et al. 2021); self-efficacy to carrying out protective behaviors or to avoid infection (Barrett and Cheung 2021; Hsing et al. 2021; Müller and Rau 2020); social norms for carrying out protective behaviors (Barrett and Cheung 2021; Müller and Rau 2020); perceived barriers to comply (White et al. 2022; Barrett and Cheung 2021; Hsing et al. 2021); perceived benefits (Barrett and Cheung 2021; Hsing et al. 2021; Müller and Rau 2020)); and knowledge/information about COVID-19 and protective measures (Barrett and Cheung 2021; Müller and Rau 2020; Luo et al. 2021).

The role of personal values was also addressed, mainly: political affiliation (e.g., voting liberal or conservative) and moral foundations (e.g., fairness, valuing group loyalty and respect for authority) (Hunt et al. 2022); motivation for protection (self and others) and pro-social attitudes (Coroiu et al. 2020); social responsibility and economic risk preferences (Müller and Rau 2020); trust in government, general trust in others (Müller and Rau 2020; Nivette et al. 2021); and police legitimacy (Nivette et al. 2021).

#### 3.3.1. Mask-Wearing

Regarding mask-wearing, Luo's study included this behavior, but did not analyze it apiece. Masks compliance was positively associated with fear of COVID-19 (Hunt et al. 2022), perceived COVID-19 severity (White et al. 2022), and perceived masking benefits and efficacy (White et al. 2022). Fairness was the only moral value positively associated with mask behavior (Hunt et al. 2022).

On the contrary, being more politically conservative, valuing group loyalty, respecting authority (Hunt et al. 2022), and perceived masking barriers (White et al. 2022) were

negatively associated with masking behavior. Perceived susceptibility and the cues to action were not associated with mask-wearing (White et al. 2022).

3.3.2. Social Distancing

Social distancing was assessed through questions that focused on different behaviors and contexts, such as avoiding groups, crowds, attending mass, or gathering (Luo et al. 2021; Nivette et al. 2021; Müller and Rau 2020); staying at home when possible infected with COVID-19 (Barrett and Cheung 2021; Nivette et al. 2021; Müller and Rau 2020); avoiding contact with sick people (Barrett and Cheung 2021; Luo et al. 2021; Nivette et al. 2021); restricting physical contact with other people; not shaking hands (Nivette et al. 2021); limiting outdoor activities (Luo et al. 2021); and using public transports when strictly necessary (Nivette et al. 2021).

Beliefs about fear, severity, or risk of COVID-19 were associated with social distance behaviors in two studies (White et al. 2022; Hsing et al. 2021), which was not confirmed by Barrett and Cheung (2021). Luo et al. (2021) did not identify predictors of social distancing adherence for the 18–24 years group specifically, but found that perceived severity of COVID-19 moderates the direct relation between age and compliance with social distance. Perceived susceptibility of COVID-19 was not identified as a predictor of social distancing (White et al. 2022).

Barrett and Cheung (2021) found a relation between perceived benefits and social distancing, which was not confirmed in another study (Hsing et al. 2021). Concerning barriers perception for social distancing, one study presents a negative association (White et al. 2022), while two others found no significant relation (Barrett and Cheung 2021; Hsing et al. 2021).

Self-efficacy for carrying out preventive behaviors was addressed in two studies (Hsing et al. 2021; Barrett and Cheung 2021) presenting opposite results. Barrett and Cheung (2021) found a positive correlation between self-efficacy for infection avoidance and distancing behavior, and identify self-efficacy as a major predictor of the adherence. Hsing et al. (2021) did not confirm this relationship.

The role of social norms to comply with physical distancing was only approached in one study (Barrett and Cheung 2021) with no significant results.

Concerning personal values, factors directly associated or predicting non-adherence to social distancing were low perception of police legitimacy, low trust in government (Nivette et al. 2021), and high economic risk tolerance (Müller and Rau 2020). Low trust in others was also related to non-adherence in one study (Nivette et al. 2021), but this result was not confirmed by Müller and Rau (2020). People who have high social responsibility tend to comply with social distancing (Müller and Rau 2020), which is in line with the results of Coroiu et al. (2020) who identified motivation for protection (self and others) and pro-social attitudes as predictors of social distancing.

Knowledge about COVID-19 was examined in two studies (Barrett and Cheung 2021; Luo et al. 2021). One of the studies (Luo et al. 2021) presented results on the impact of information on adherence to preventive actions for all age groups combined, having no specific results for young adults. The other study did not confirm a relationship between information and adherence to social distancing.

## 4. Discussion

Several measures were taken to mitigate the COVID-19 pandemic, namely social distancing and mask use. The efficacy of these measures alone (Kinyili et al. 2022; Sun et al. 2022) or combined (Rao et al. 2021) to reduce the Reproduction Rate (Ro) of COVID-19 is well established, reinforcing the importance of adherence to these behaviors for personal and community protection.

Due to the absence of the previous literature reviews on the cognitive determinants of adherence to social distancing and mask use in young adults, the discussion will

address the main cognitive determinants for this age group and compare the results with other reviews from the general population.

Seven of the eight studies under analysis were cross-sectional and all have a non-probability samples, which is similar to the results reported by previous reviews on COVID-19 protective behaviors (Minozzi et al. 2021; Noone et al. 2021).

The cognitive determinants that were under study can be integrated with several cognitive theoretical models: self-efficacy (in SCT); perceived benefits and barriers to comply (in SCT; HBM; TPB); fear/severity/risk of COVID-19 and susceptibility (in HBM); and social norms for carrying out protective behaviors (in TBP). Self-control, which is a relevant determinant of health behaviors and is considered in other studies on adherence to COVID-19 protective behaviors (e.g., Bieleke et al. 2023; Rodriguez et al. 2023), was not included in any of the studies under analysis. Along with these determinants, knowledge about COVID-19 was considered in other studies (Barrett and Cheung 2021) and personal values were addressed in four studies (Hunt et al. 2022; Coroiu et al. 2020; Müller and Rau 2020; Nivette et al. 2021).

### 4.1. Social Distancing

Concerning the assessment of social distancing, all studies under analysis used questionnaires. This is partially in line with the scoping review by Noone et al. (2021), which reports that 63% of the studies used self-report measures of social distancing, while 37% relied on smartphone GPS data to quantify mobility. The questionnaires focused on various behaviors for assessing social distance adherence (e.g., avoiding crowds and gatherings, staying at home, avoiding contact, limiting outdoor activities, and avoiding contact with infected people). This may justify some of the heterogeneity found in the results of the present review.

Risk perception is vital in leading people to achieve appropriate health behavior (Brewer et al. 2007). Two of three studies under analysis conclude that beliefs about fear, severity, or risk of COVID-19 have a positive effect on social distance adherence in young people. For the general population, this effect is well established (Cipolletta et al. 2022; Sadjadi et al. 2021). Some heterogeneity in our results can indicate that risk perception in young adults does not necessarily imply adherence to social distancing, in contrast to older people. Although this age group seems to be more reluctant to adopt COVID-19 protective behavior (Haischer et al. 2020; Kim and Crimmins 2020), one of the studies under analysis (Luo et al. 2021) concludes that risk perception moderates the effect of age on the adherence to social distancing.

Perceived susceptibility (perceived likelihood of being infected and/or suffering serious health consequences due to COVID-19) was not confirmed as a predictor of social distancing in young adults. These conclusions do not confirm the results for general population in the review by Urbán et al. (2021), where perceived severity was found to be a stronger predictor of adherence rate to preventive behaviors, along with perceived susceptibility, across several countries. However, our results were aligned with studies developed in Western countries (Liang et al. 2022), showing that those who assessed the infection as more severe were more willing to adopt preventative health practices, such as social distancing.

The results of the two studies (Barrett and Cheung 2021; Hsing et al. 2021) that focus on the impact of self-efficacy in social distancing behavior are contradictory. Barrett and Cheung found that self-efficacy was a major predictor of social distancing in young adults. However, they combine two types of self-efficacy (for infection avoidance and distancing behavior). The study by Hsing et al. (2021), which discusses the impact of self-efficacy on the carrying out of social distancing, found no significant results.

Social norms (i.e., individuals' perceptions of the behavior of others or injunctive norms that involve others' attitudes or opinions regarding a behavior) can drive and change healthier perceptions and behaviors (Mattern and Neighbors 2004). Nevertheless,

the only study in the present review that analyses the impact on social norms does not confirm this relation for physical distancing in young adults (Barrett and Cheung 2021).

Although the impact of conspiracy beliefs has been widely studied in the literature on the determinants of adherence to COVID-19 preventive measures (Bierwiaczonek et al. 2022), no article under analysis has focused on this belief in young people.

Based in only one study (Barrett and Cheung 2021), knowledge about COVID-19 seems to not be related to the adherence of young people to social distancing. That is not surprising, since it is well known that information is a necessary but insufficient requirement for changing health behaviors (Kelly and Barker 2016). Nevertheless, it is important to remember that knowledge acquisition depends of the confidence in the information source. For instance, Fridman et al. (2020) concluded that young Americans express higher trust in private sources (e.g., private TV networks) and social networks (e.g., Twitter) compared to governmental sources.

A set of beliefs related to moral values were considered in this review. When considering moral values, it is important to bear in mind that the degree of adherence to certain values is not fixed and has been found to fluctuate during the pandemic. A study with the French population (Bonetto et al. 2021) found that conservation values (favoring stability and preserving traditional practices) were higher than normal during the outbreak of COVID-19 and was strongly related to adherence to social distancing, while self-enhancement (to favor personal interests to the detriment of those of others) and openness to change (orientation towards change and independence) values were lower during the same period. Also based on the four value domains proposed by Schwartz (self-transcendence, self-enhancement, conservation, and openness to change), Potocan and Nedelko (2023) concluded that openness to change and self-enhancement values decreased more during the pandemic, and conservation and self-transcendence (transcending self-interest to promote the well-being of other values) decreased less in a sample of young adults in Slovenia (Bonetto et al. 2021). In our review, results are scattered among the articles under analysis, since each moral value was addressed only by one study, except for social responsibility, which was confirmed as a predictor of social distancing in two studies (Coroiu et al. 2020; Müller and Rau 2020). This relevance of pro-social values is not confirmed by Bonetto et al. (2021), who identified that only conservation values (favoring stability and preserving traditional practices) were a determinant of adherence to social distance. Our results seem to confirm that low trust in the government and in the police (Nivette et al. 2021) is related to non-adherence to social distancing. Although trust in the police is a relatively understudied determinant (mainly related to trust in authority), trust in the government is identified as a predictor of social distancing for the general population (e.g., Fridman et al. 2020).

Although Müller and Rau (2020) addressed economic risk tolerance, none of the studies under focus on COVID-19 risk tolerance, which seems to influence adherence to physical distance. Sheth and Wright (2020) identify an inverse relationship between risk tolerance and adherence to social distancing when obtaining basic services, but not for attending work or social interactions. Although this conclusion refers to the general population, it points out different impacts of risk tolerance depending on contexts, particularly the diminished impact of risk tolerance when it comes to social interactions, a frequent behavior in young adults.

*4.2. Mask-Wearing*

Only three studies included in this systematic review analyze the use of masks. In addition, the study by Luo investigated several COVID-19 prevention behaviors, and it was impossible to isolate adherence to mask use, since the authors studied adherence in general and not specific behaviors. The two included studies used self-report questionnaires for data collection, with the White et al. (2022) study attempting to identify HBM constructs related to mask-wearing and the Hunt et al. (2022) study examining the correlates of core values (e.g., moral values of fairness, responsibility, loyalty, and respect for

authority) and social influence on mask non-compliance in undergraduates at a selective American university.

Perceived COVID-19 severity, masking benefits, barriers, and self-efficacy were the HBM constructs associated with masking behavior. Earlier meta-analysis (Carpenter 2010) identified benefits and barriers as the strongest predictors of HBM's Benefits. As Shelus et al. (2020) found, in a qualitative study with focus groups, youth adults perceived masks as a self-protection and respect for others behavior, but they also considered it physically and social discomfortable. The perceived COVID-19 severity and susceptibility in association with mask-wearing was not found in previous studies (Liang et al. 2022).

The finding that perceived susceptibility was not linked to masking behavior is consistent with earlier meta-analyses (Carpenter 2010) related to the effectiveness of HBM in predicting behavior, which identified susceptibility as the HBM's weakest predictor, but not with Liang et al. (2022) systematic review and meta-analysis, who found that the associations between perceived susceptibility and facemask wearing were significant in Western samples. However, it should be mentioned that political inclination could influence mask-wearing in the USA.

Hunt et al. (2022) present a regression predicting the face-wearing model, which includes political orientation (i.e., liberal or conservative) and three moral values, namely loyalty, fairness, and respect for authority, with the latest (i.e., respect for authority) being the strongest predictor.

Only fairness was associated with mask compliance. The other values and being conservative (political orientation) were associated with non-mask use. Regarding political orientation, a comprehensive review by Shushtari et al. (2021) shows that political inclinations influenced mask-wearing, with Democrats more likely to use it than Republicans. Relating to personal values, there are several studies (e.g., Bonetto et al. 2021; Vecchione 2022) that analyze the relationship between these and prevention behaviors; however, most use the Portrait Values Questionnaire, and therefore comparisons with the moral values of Hunt et al. (2022) study are not feasible.

Some limitations must be considered regarding the present systematic review. Despite the importance of young people's adherence to COVID-19 protective measures, we found remarkably few quantitative studies for this group focusing specifically on cognitive determinants of social distancing and mask use. More valuable studies may have been published at the time of this research. In addition, although the researchers attempted to conduct the best possible analysis using the available studies, the exclusive use of self-report measures in the eight studies under analysis could represent an overestimation of adherence to COVID-19 preventive behavior (Davies et al. 2022), generating some bias in the results.

## 5. Conclusions

The analysis carried out in this study allowed for the summarization of individual cognitive factors that contributed to young adults' highly recommended preventive social behavior (distancing and/or the use of facial masks) to reduce the spread of COVID-19.

The results reinforce the assertions of cognitive models, highlighting the relevance of cognitive constructs in explaining adherence behaviors to preventive measures.

Similar to previous studies, higher levels of risk perception (i.e., the probability that one will be harmed if nothing is done) and higher degree of perceived severity (i.e., the potential harm of COVID) were significantly related to better outcomes in adherence both to social distancing and the use of masks. However, perceived susceptibility (i.e., beliefs about the vulnerability of being infected and/or having a severe form of COVID-19) did not show an effect, which may result from invulnerability beliefs that are very common in young adults.

Perceived self-efficacy in preventing contamination was identify as predictor of social distancing when self-efficacy for infection avoidance and for distancing behaviors were combined.

Other factors that have been demonstrated to predict both the adherence behavior to the use of mask and social distancing include moral values, particularly fairness (i.e., defending equal opportunities and benefits), suggesting that young individuals may exhibit greater willingness to adhere when they perceive themselves as socially responsible, believing that their behavior is shared by the community and has positive consequences for it; and trust in the government (i.e., perceived quality of institutional communication and respect for authority), which gives governments and official institutions a great responsibility when thinking about communication to promote adherence to their recommendations regarding preventive behaviors.

The knowledge of these individual and community cognitive beliefs allows for the development of adequate methods of intervention, centered on the people to whom the message is intended to reach. Moreover, effective communication is crucial, whether conveying scientific information or disseminating details about political measures. Such communication plays a pivotal role in fostering confidence among populations residing at a distance from central political and/or scientific contexts. This underscores the significance of adopting creative and innovative approaches to sensitize young adults to adhere to specific public health requirements and best good practices.

**Supplementary Materials:** The following supporting information can be downloaded at: https://www.mdpi.com/article/10.3390/socsci13050275/s1, Figure S1: Quality assessment based on JBI Critical Appraisal Checklist for Cross Sectional Studies; and Table S1: Quality assessment based on JBI Critical Appraisal Checklist for Cohort Studies.

**Author Contributions:** Conceptualization, G.A. and M.S.; methodology, M.A.-S. and G.A.; validation, M.A.-S. and G.A.; formal analysis, T.L. and A.G.; writing—original draft preparation, M.A.-S. and T.L.; writing—review and editing, G.A., A.G. and M.A.-S. All authors have read and agreed to the published version of the manuscript.

**Funding:** This work was supported by the project "Youth Breakdown in the post-COVID era and their Vaccination Intention" (IPL/2021/Vaccin2You(th)_ESTeSL), funded by Instituto Politécnico de Lisboa, Portugal.

**Institutional Review Board Statement:** Not applicable.

**Informed Consent Statement:** Not applicable.

**Data Availability Statement:** Data sharing is not applicable since no new data were created or analyzed in this study. All data are contained within the article.

**Acknowledgements:** This work was supported by the project "Youth Breakdown in the post-COVID era and their Vaccination Intention" (IPL/2021/Vaccin2You(th)_ESTeSL), funded by Instituto Politécnico de Lisboa, Portugal. The H&TRC authors also acknowledge to FCT/MCTES for the funding by UIDP/05608/2020 (https://doi.org/10.54499/UIDP/05608/2020) and UIDB/05608/2020 (https://doi.org/10.54499/UIDB/05608/2020).

**Conflicts of Interest:** The authors declare no conflict of interest.

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
