# Peer review of "Influence of Cognitive Factors on Adherence to Social Distancing and the Use of Masks during the COVID-19 Pandemic by Young Adults: A Systematic Review"

_socsci, doi:10.3390/socsci13050275_

Round 1

Reviewer 1 Report

Comments and Suggestions for Authors

The fundamental principle underlying the classification of a pursuit as science lies in its replicability, without which the very essence of scientific inquiry is compromised. Thus, I tried to replicate your PubMed search, using your own conditions and after adding time frame 2020-2022 (in bracket): 

First condition: 97 (83) articles

Second condition: 46,125 (10,313)  articles

Third condition: 16 (13) articles

Fourth condition: 20 (19) articles 

As per the intersection of sets, following your stated logic of “#1 AND #2 AND #3 AND #4,” I obtained 0 articles. However, in Figure 2, you claimed to have retrieved 373 articles from PubMed. This discrepancy is significant, and upon careful review, I couldn't identify any typos that might explain this discrepancy. Notably, taking the union of sets would result in 46k (10k) articles. If you had included the third or fourth condition in any intersection, the total should not have exceeded those values.

In conclusion, if your research primarily involves a database search, a complete failure in executing this search undermines the validity of the entire body of work built upon it.

Author Response

Herewith we enclosed the revised manuscript to be submitted to the Social Science, entitled "Influence of cognitive factors on adherence to social distancing and the use of mask during the COVID-19 pandemic by young adults: a systematic review".

We are grateful for the helpful comments and questions raised by the reviewers. We have attended to these issues above, hoping our answers adequately address the reviewers’ comments. All the modifications in the manuscript are highlighted in track changes and coloured in red and yellow.

We thank the reviewers' contribution, which helped improve the manuscript’s quality. We hope this version complies with the Social Science standards for publication.

Sincerely yours,

Marina Almeida-Silva

Reviewer 2 Report

Comments and Suggestions for Authors

This systemic review addresses barriers to the use of preventive measures during COVID-19 in young adults. This is an important issue for health messaging and understanding what might have been done better during COVID to improve future compliance with recommendations during pandemics.

 The paper needs English editing, mostly for punctuation.  Aside from grammatical issues, it is clearly written and relatively organized.  It seems as if there is a larger background on the social cognitive model, TBP and HBM than is needed.  Few of the include studies used these models. There was one cognitive model and one HBM, but no obvious TBP papers.  This material may not add much to the systematic review itself which was more practice-based with fewer conceptual models employed.

 The role of the background information on these conceptual models of behavior change does not seem to provide a strong framework in the reporting of the results, although is a bit clearer in the discussion.  I think it would be helpful to focus on the aspects of these models related to masking and distancing from the 8 studies.  Even though many of the studies did not use a conceptual model, identifying aspects that fit into a conceptual finding and then comparing across studies would be easier  to understand.  The way it is written presently, it is not entirely clear what the purpose of the systematic review is.  I think the purpose should be clearly stated in terms of the conceptual models if these are the focus of the review.

 Other comments:

 Line 20: Analyze should be analysis

Line 36: Nonexistence rather than inexistence?

Line 81: theory TBP is redundant

Line 82: extra “it”

 First figure on page 9 does not have a figure number and is not referenced in the text. Also, there are seven studies for which quality data is provided, but the text states that 8 studies were included.

 For what reasons were the 1,946 reports excluded in the diagram on page 11. How was relevance determined? This seems like a plethora given the specific search terms used. It seems as if many of these studies were excluded because they did not measure the cognitive constructs related to the social cognitive model, TPB, or the BHM.  Is that how the exclusions were made?

 It would be helpful in 3.3 Outcomes to have subheadings for each outcome.

Comments on the Quality of English Language

The English language usage needs some work. The problems are mostly grammatical and due to punctuation errors.

Author Response

(The authors gave the same response as above.)

Reviewer 3 Report

Comments and Suggestions for Authors

This study aims to systematise the scientific evidence related to cognitive factors that underlie the intention of young adults' intention to adhere to preventive social behaviour (distancing and/or the use of facial masks) against COVID-19.

The topic of the manuscript is within the scope of the Journal. I find that manuscript is relevant to the aims of Social Sciences.

 TITLE

I think that the title of the article is accurate.

 ABSTRACT

The Abstract should be rewritten to include the PEO (Population, Exposure, and Outcome) statement for this review.

 INTRODUCTION

The introduction creates a niche for the current investigation by showing a hole in the literature and discussing how it intends to fill it. Some clarifications are however needed. I think, the source used to define the young adults aged 18-35 should be provided.

 METHOD

Some clarifications are needed. I suppose that age range of young adults must not only defined but also explained (the source used to define the young adults aged 18-35 should be provided).

 RESULTS

Some clarifications are needed.

Why is written in the Table 4 that Barret and Cheung, 2020 instead of Barret and Cheung, 2021?

Why is written in the Table 4 that age ranges of participants were defined from 18 to 52 years, from 18 to 44 years, despite definition of the young adults (aged 18-35)?

. 

DISCUSSION

If these upper age ranges for young people were not strictly defined in the selected publications, this should also be noted as a limitation.

 TO SUM UP I think the author(s) need to make the recommended corrections.

Author Response

(The authors gave the same response as above.)

Reviewer 4 Report

Comments and Suggestions for Authors

The paper is of great interest and is well structured.

Abstract: The abstract provides a clear overview of the study's objectives, methods, and key findings.

Introduction: The introduction effectively sets the stage for the study and highlights. The authors identify the problem and formulate a very well defined question for the problem at hand. The different social cognitive models are well explained and commented by the authors.

The authors introduce the reader conveniently through the introduction.

Methods: This section is exhaustive and well described.

However, the selection of papers should be explained in greater depth. The number of studies excluded by each of the exclusion criteria must be explained.

Page 11 of 27. The inclusion criteria are well defined, however, when they refer to Young adults they include subjects aged 18-35 years, but later they refer to 18-39 years (page 11 of 27 last paragraph).

The inclusion criteria include European, Canadian and North American studies however one of the included studies was carried out with United States, but also with Mexico, China and Taiwan population.

In the Flow diagram the authors point out that 3171 studies have been registered and 1042 have been removed before screening, the cause is duplicate records. It is due to the duplication between the different databases. It should be explained conveniently (page 10 of 27).

Results: The study by Hunt et al 2022 analyses variables such as heat anxiety, political affiliation and moral foundation. These issues must be adequately explained.

 Discussion:  It is well structured. There is some error that needs to be corrected. For example, the word 'percetion' for perception.

On page 23 of 27, in the fifth paragraph, the fluctuation during the pandemic of the degree of adherence to certain values should be conveniently explained.

Also, in the seventh paragraph the phrase 'several Covid 19 prevention behaviours, and it was impossible to isolate adherence to mask use.

In page 24 of 27 second paragraph it should be mentioned that the political inclination influenced mask-wearing in the USA.

The word republicans should be lowercase like democrats. Or at least both must be either uppercase or lowercase.

There is an error in the last line of the ‘behaviorset’ discussion paragraph.

Possible observer biases should also be discussed, as occurs in any systematic review.

 Conclusion: Some of the sentences are long and complex, which may make it challenging for readers to follow. Simplifying the language and breaking down complex ideas into smaller sentences would improve readability.

References: The references must be exposed in a uniform way, so they must be reviewed.

Comments on the Quality of English Language

The English language is correct. Some errors need to be checked. 

Author Response

(The authors gave the same response as above.)

Round 2

Reviewer 1 Report

Comments and Suggestions for Authors

I'd say that the queries provided in the updated version are within the realm of possibility. While they appear somewhat sloppy (this condition could have been expressed in a much more elegant and less repetitive way), let them be. However, assuming that the key point of your study is conducting such formalized search, I'm somewhat perplexed why you even removed the figure showing disaggregated database results.

Let's get it right.

The step where you first gather a large number of papers and then eliminate the overwhelming majority immediately is completely unclear. Show the base and how many papers were retrieved. Since you prefer to conduct four searches per database, show the number each search yielded and how many were immediately eliminated as duplicate query results from the same database. Then clearly illustrate how many you eliminated in the next step as overlapping between bases.

Author Response

Herewith we enclosed the revised manuscript to be submitted to the Social Science, entitled "Influence of cognitive factors on adherence to social distancing and the use of mask during the COVID-19 pandemic by young adults: a systematic review".
We are grateful for the helpful comments and questions raised by the reviewers. We have attended to these issues above, hoping our answers adequately address the reviewers’ comments. All the modifications in the manuscript are highlighted in track changes and coloured in red and yellow.

We kindly thank the reviewers' contribution, which helped improve the manuscript’s quality. We hope this version complies with the Social Science standards for publication.

Sincerely yours,

Reviewer 2 Report

Comments and Suggestions for Authors

The paper requires extensive, but somewhat minor, English editing.  The Bandura quote on lines 58-59 must be incorrect because the grammar is incorrect.

The paper is based around the three behavioral models, TPB, HBM, and SCT, but the organization is based on mask-wearing and social distancing. It would be good to come to a conclusion about what conceptual model appears to best fit the COVID-19 data examined. Due to the many layers beyond personal beliefs, the SCT seems more relevant due to the political and cultural influences on decision making. I think the paper would be easier to understand if the organization better aligned with the purpose.  Is the paper about these conceptual models and whether they are useful in explaining COVID-19 protective behaviors or about what predicts social distancing and mask wearing?

Comments on the Quality of English Language

English editing is necessary throughout.  

Author Response

(The authors gave the same response as above.)

Round 3

Reviewer 1 Report

Comments and Suggestions for Authors

I reiterate the issue, as it has not been addressed at all. According to your search methodology, you make four queries per database. This practice is somewhat unusual. However, as you prefer it, then please proceed and be specific about the results you obtained. Show the overlap per database and indicate how many entries were eliminated at this step. Then demonstrate how much overlap was removed after merging the results from your databases. All such information should be included in your Figure 2. Otherwise, it serves merely decorative purposes and fails to convey relevant scientific information about your search criteria.

Author Response

Dear reviewer,

Your comments were always taken into consideration and improved a lot our manuscript.

As described in the document, we used the Rayan Platform to deal with the massive data provided by each database. Saying so, the software identify automatically the duplicates and erase them from the platform. This means that we cannot trace from which database those erased files were provided. Moreover, we merged the downloaded files in one ZIP folder. Perhaps, this was not a good approach, but it's the reason why we couldn't show the overlap per database and indicate how many entries were eliminated at this step.

Besides this constrain, the authors believe on the accurate results provided by this work.

Thank you again for your comments.